# Sources of successful participant engagement in a public health research study: A focus on a Latino community

Angel Lomeli[1,2*], Arleth A. Escoto[1], Breanna Reyes[1], Kayleigh Kornher[3], Keira Beltran-Murillo[3], Kathia Nuñez[3], Ariel Cohen[3], Maria Linda Burola[1], Isabel Villegas[3], Scarlet Flores[4], Ana Perez-Portillo[4], Norma Porras[4], Melody Torres[3], Linda Salgin[4], Kelli L. Cain[2], Nicole A. Stadnick[5,6,7], Louise C. Laurent[1], Borsika A. Rabin[2,7], Marva Seifert[8]

1 Department of Obstetrics, Gynecology, and Reproductive Sciences, University of California San Diego, La Jolla, California, United States of America, 2 Herbert Wertheim School of Public Health and Human Longevity Science, University of California San Diego, La Jolla, California, United States of America, 3 University of California San Diego, La Jolla, California, United States of America, 4 Department of Research and Health Promotion, San Ysidro Health, San Ysidro, California, United States of America, 5 Department of Psychiatry, University of California San Diego, La Jolla, California, United States of America, 6 Child and Adolescent Services Research Center, San Diego, California, United States of America, 7 Altman Clinical and Translational Research Institute Dissemination and Implementation Science Center, University of California San Diego, La Jolla, California, United States of America, 8 Department of Medicine, University of California San Diego, La Jolla, California, United States of America

* aalomeli@ucsd.edu

## Abstract

### Background

Latino populations remain vastly underrepresented in clinical and translational research. This study aims to characterize the most common sources of successful participant engagement within our sample.

### Methods

Between February 2022 and March 2023, research staff systematically recorded how participants learned about an ongoing study (which we term source of successful participant engagement) designed to co-create and implement a COVID-19 testing program in a U.S./Mexico border community. Demographic characteristics were correlated with each source of participant engagement at the univariate level using a chi-squared test and, if significant, were included in a multinomial logistic regression model to determine the association between participant characteristics and source of participant engagement.

**Data availability statement:** All relevant data are within the manuscript and its Supporting Information files.

**Funding:** This study was funded by NIH/NIEHS P42ES010337-20S1, awarded to Drs. Robert Tukey and Louise Laurent. The funders had no role in the design and conduct of the study; collection, management, analysis, and interpretation of the data; preparation, review, or approval of the manuscript; or decision to submit the manuscript for publication.

**Competing interests:** The authors have declared that no competing interests exist.

## Results

A total of 2836 individuals responded to questions regarding source of participant engagement; the most common responses were: Word of Mouth (32%), Clinic/Provider referral (32%), and Walk Up to the testing site (21%). Males were 35% less likely than female participants to report having heard of the study through their Clinic/Provider compared to Walk Up (p < .01). Participants <18 years of age were 2.78 times as likely compared to individuals >54 years of age to have learned about the study through Word of Mouth compared to Walk Up (p < .01). Compared to Walk Up, participants who lived outside San Ysidro were 2.36 times more likely to be recruited through their Clinic/Provider (p < .01) and 2.11 times more likely through Word of Mouth (p < .01), compared those in San Ysidro. Education and clinical symptoms were not significantly associated with engagement source.

## Conclusion

Advancing our understanding of sources of successful participant engagement in marginalized communities is necessary to increase equitable participation in clinical and translational research.

## Introduction

Latino populations remain vastly underrepresented in clinical and translational research [1]. Non-Hispanic White Americans comprise 58.4% of the total population yet studies have found that they make up more than 90% of participants in research studies [1–2]. Multiple barriers prevent underserved communities from participating in research studies, including perceived stigma and discrimination, social status, language barriers, misinformation, lack of knowledge and understanding about the research process, legal status, and fear of deportation [3]. As a result, participants from these communities may be reluctant to engage in research studies, compromising both the validity and generalizability of research findings [4]. Populations who are non-English speakers, on average, have a lower educational background, lower socioeconomic status, and work in high-risk occupations, making them more likely to experience worse health outcomes [5]. The inadvertent and systematic exclusion of these vulnerable populations from research not only affects the allocation of health services in underserved communities, further exacerbating poor health outcomes, but also undermines health equity and perpetuates systemic injustices by failing to address the unique needs of these communities, further justifying the need to fully understand the root causes of the disproportionality of representation in research [6].

The COVID-19 pandemic amplified the burden of disease and other social determinants of health among racial and ethnic minority groups [7,8]. For instance, the increased likelihood of being unable to work from home and residing in crowded multi-generational households imposed greater risks for infection and

transmission [9]. Ethnic minority groups were also more likely to be hospitalized due to COVID-19 infection compared to non-Hispanic White patients [9,10]. Community-based research has become both an ethical and statistical necessity for understanding healthcare access across diverse populations [11,12]. Located in Southern California, in the county of San Diego, the border city of San Ysidro (SY), a predominantly Latino community where 83% identify as Hispanic/Latino(a), was selected for this study because its population is often underrepresented in research, despite its location at one of the world's busiest international border crossings and proximity to Tijuana, Mexico. Members of this community are more likely to be exposed to adverse living and social situations associated with poor health outcomes, such as substandard housing and underemployment, with about a third of household incomes falling under $30,000 a year [13–15]. Furthermore, *Promotores,* known as Spanish-speaking Community Health Workers who help bridge patient care in a culturally relevant manner, play a key role in building trust between residents and healthcare administrators while providing culturally appropriate information [15,16]. In our study, we partnered with these Promotores to recruit participants, leveraging their established relationships and deep understanding of the community. Through the translation of existing knowledge regarding the effectiveness of the use of Promotores, this study sought to characterize sources of participant engagement with the Latino community living among the U.S./Mexico border region. Between February 2022 and March 2023, research staff systematically collected and evaluated the self-reported ways individuals learned about the study. In this paper, we refer to this as a source of participant engagement.

## Methods

### Study design

Community-driven Optimization of COVID-19 testing to Reach and Engage underserved Areas for Testing Equity, in Women and Children (CO-CREATE) was a two-year research study funded by the National Institutes of Health (NIH) Rapid Acceleration of Diagnostics for Underserved Populations (RADx-UP) program designed to mitigate disparities experienced by underserved communities by offering no-cost COVID-19 polymerase chain reaction (PCR) testing for children, pregnant persons and their families, and the community. The study sought to understand the most prominent barriers and facilitators for COVID-19 testing and access among traditionally underserved communities during the pandemic. Using that knowledge, the team worked with a community and scientific advisory board to co-create a culturally appropriate COVID-19 testing program. In order to achieve this goal, University of California San Diego (UCSD) partnered with a Federally Qualified Health Center (FQHC) in San Diego and a social change organization, to develop and implement a COVID-19 testing program in the San Ysidro community, one of the most affected areas by COVID-19 in San Diego County [13]. The testing site was located in the parking lot of the FQHC. This was advertised as a no-cost COVID-19 testing site first where patients and community members were invited into the study after testing; consenting and participation in the research study was not a requirement to receive a no-cost COVID-19 test. Implementation and outcomes of the CO-CREATE study have been previously described [13–15]. For this analysis, data collected between February 1, 2022 and March 31, 2023 were used.

### Study procedures and variables of interest

The CO-CREATE project offered no-cost, walk-up COVID-19 testing to all community members. The eligibility criteria for enrollment in the study included the ability to provide informed consent, either by the individual or their guardian. There were no other eligibility requirements. Individuals who had already been tested for COVID-19 through the CO-CREATE project were marked as "return" and omitted from this analysis. Individuals who registered to test for COVID-19 were also invited to complete a survey about their demographic and health characteristics and received a 20 USD incentive. Participant characteristics and demographic factors used in the analyses were drawn from this survey. Those who declined to participate in the research study were still eligible to receive a no-cost COVID-19 test.

Study staff asked each person who requested a COVID-19 test how they heard about the CO-CREATE research study. Participant responses were open-ended (i.e., they were not offered a forced choice response menu). Research staff documented and organized each response into different categories. A group of 3–5 study staff regularly met after all of the data were collected to combine free-response answers into categories. Discrepancies were discussed among the team and, if necessary, voted on. The final engagement categories were:

1. **Clinic/Provider Referral:** Participants learned about our testing site from clinic staff or their provider.

2. **Community Center:** Participants were informed about our testing site through a local community center.

3. **County Helpline:** Participants were informed about our testing site through one of the San Diego County helplines, such as 211.

4. **Flyer:** Participants came across our printed flyer, which was displayed in public locations in the San Ysidro area, informing them about our testing site. Some locations included public parks, public libraries, local grocery stores, and local post offices.

5. **Highway Billboard:** Participants learned about our testing site by seeing one of our displayed highway billboards. Four billboards facing the highway were used for approximately five weeks.

6. **Insurance:** Participants were informed about our testing site through their health insurance.

7. **On-site Recruitment:** Participants were recruited in person within our testing site by study staff, who approached patients in waiting rooms to inform them about our study.

8. **Online – Internet Search:** Participants were looking for a COVID-19 testing site online and came across the CO-CREATE project.

9. **Online – Social Media:** Participants found out about our testing site through any form of social media.

10. **School:** Participants were informed about our testing site through their child's school.

11. **Walk Up:** Participants saw our testing site and walked up to it.

12. **Word of Mouth:** Participants were informed about our testing site through personal contacts, such as friends or family members, not including work, school, or insurance.

13. **Work:** Participant was informed about our testing site through their work.

Study staff documented the source of successful participant study engagement in each new participant's profile. Paid advertising campaigns included digital bus stop signs, local radio segments, and highway billboards. Notably, only the highway billboard recruited a participant, while the other advertising campaigns did not. The frequency of each source of participant engagement is shown in Fig 1.

Gathered from the post-testing survey, participants self-reported sex, ethnicity, age, city, education, and clinical symptoms.

For the symptoms variable, participants were given a list of symptoms in a matrix-style question including fever or chills, cough, shortness of breath or difficulty breathing, lack of energy or general tired feeling, muscle or body aches, headache, new loss of taste or smell, sore throat, congestion or runny nose, feeling sick to your stomach or vomiting, diarrhea, abdominal pain, skin sash, and other (participant would be prompted to fill out "other" if selected). For the purpose of this study, if a participant answered "yes" to at least one of the aforementioned symptoms, they were classified as "symptomatic".

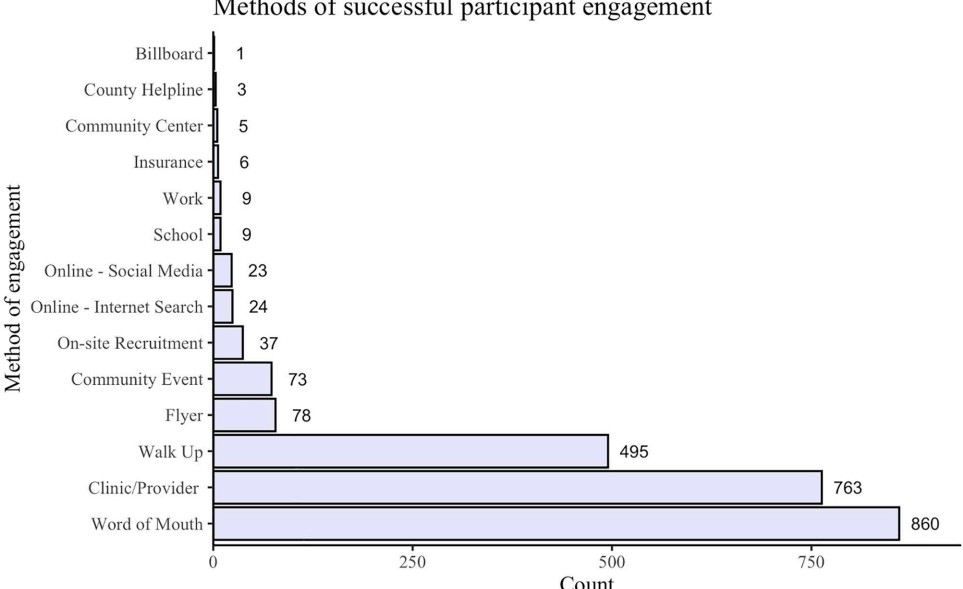

**Fig 1. Frequency of self-reported source of participant engagement (N = 2386).**

## Ethics

This study received approval from the Institutional Review Board at the University of California San Diego Human Research Protections Program, protocol number 210498. All participants provided informed verbal consent or informed verbal assent from a parent or guardian when appropriate, acknowledging potential risks and the voluntary nature of study participation. Upon participants providing consent or assent, a witness study staff member signed a paper copy of the consent/assent, and the participant was provided a copy of the consent/assent form for their records. Participants who did not provide any form of consent or assent are not part of this study, even if they provided information regarding their source of participant engagement.

## Statistical analysis

Descriptive statistics were used to characterize participant demographics and chi-squared tests were used to compare participant demographics among the top three sources of successful participant engagement (Walk Up, Clinic/Provider Referral, and Word of Mouth). A subsequent multinomial logistic regression model was fit to include statistically significant (at $p < .05$) demographic variables. Walk Up is the outcome reference category for all comparisons.

## Results

### Demographics

Between February 2022 and March 2023, 2386 unique participants reported their primary source of engagement. Out of those participants, 860 (36%) were recruited through Word of Mouth, 763 (32%) through a referral from their Clinic/Provider, and 495 (21%) by Walk Up. The other identified sources of successful participant study engagement were reported at lower frequencies. See Fig 1 for more details.

### Chi-squared analysis

The top three sources of successful participant study engagement, Walk Up, Clinic/Provider, and Word of Mouth, were analyzed through cross-tabulation and chi-squared tests for each associated factor of interest (Table 1). Among the

**Table 1. Distribution of participant demographics by top three source of successful participant engagement.**

| Variable | Walk Up - 495 (23.4%) | Clinic/Provider −763 (36.0%) | Word of Mouth − 860 (40.6%) | Total - 2118 | p |
|---|---|---|---|---|---|
| **Sex** | | | | | **<.01** |
| Female | 282 (57.0%) | 502 (65.8%) | 438 (50.9%) | 1222 (57.7%) | |
| Male | 213 (43.0%) | 261 (34.2%) | 422 (49.1%) | 896 (42.3%) | |
| **Ethnicity** | | | | | **<.01** |
| Hispanic | 465 (93.9%) | 708 (92.8%) | 736 (85.6%) | 1909 (90.1%) | |
| Not Hispanic | 30 (6.1%) | 55 (7.2%) | 124 (14.4%) | 209 (9.9%) | |
| **Age** | | | | | **<.01** |
| <18 | 48 (9.7%) | 111 (14.6%) | 128 (14.9%) | 287 (13.6%) | |
| 18-24 | 34 (6.9%) | 43 (5.6%) | 109 (12.7%) | 186 (8.8%) | |
| 25-34 | 68 (13.7%) | 131 (17.2%) | 150 (17.4%) | 349 (16.5%) | |
| 35-44 | 57 (11.5%) | 99 (13.0%) | 108 (12.6%) | 264 (12.5%) | |
| 45-54 | 93 (18.8%) | 122 (16.0%) | 120 (14.0%) | 335 (15.8%) | |
| >54 | 195 (39.4%) | 257 (33.7%) | 245 (28.5%) | 697 (32.9%) | |
| **City** | | | | | **<.01** |
| San Ysidro | 226 (45.7%) | 203 (26.6%) | 233 (27.1%) | 662 (31.3%) | |
| Outside SY | 263 (18.7%) | 541 (38.4%) | 605 (42.9%) | 1409 (66.5%) | |
| Prefer not to say | 6 (1.2%) | 19 (2.5%) | 22 (2.6%) | 47 (2.2%) | |
| **Education** | | | | | **<.01** |
| No High School | 91 (18.4%) | 141 (18.5%) | 171 (19.9%) | 403 (19.0%) | |
| High School | 117 (23.6%) | 175 (22.9%) | 250 (29.1%) | 542 (25.6%) | |
| University | 23 (4.7%) | 48 (6.3%) | 64 (7.4%) | 135 (6.4%) | |
| Prefer not to say | 264 (53.3%) | 399 (52.3%) | 375 (43.6%) | 1038 (49.0%) | |
| **Symptoms** | | | | | **<.01** |
| Asymptomatic | 283 (57.2%) | 401 (52.6%) | 547 (63.6%) | 1231 (58.1%) | |
| Symptomatic | 212 (42.8%) | 362 (47.4%) | 313 (36.4%) | 887 (41.9%) | |

respondents included in this model, 57.7% were female, 90.1% were of Hispanic/Latino(a) ethnicity, and 32.9% were 55 years of age or older. See Table 1 for more details.

## Multinomial logistic regression model

Compared to female participants, male participants were 35% less likely to report having heard of the CO-CREATE testing site through their Clinic/Provider compared to Walk Up (OR =.65, 95% CI = [.51,.83], $p < .01$). Compared to participants who self-identified as Hispanic/Latino(a), those who self-identified as non-Hispanic/Latino(a) were 2.12 times more likely to report having heard of the testing site through Word of Mouth compared to Walk Up (OR = 2.12, CI = [1.37, 3.29], $p < .01$). Participants aged <18 were 1.93 times more likely to be recruited through their Clinic/Provider (OR = 1.93, CI = [1.29, 2.88], $p < .01$) and 2.78 times more likely to report having heard of the testing site through Word of Mouth (OR = 2.78, CI = [1.87, 4.14], $p < .01$) compared to Walk Up, compared to participants aged >54. Participants aged 18–24 were 2.88 times more likely to be recruited through Word of Mouth compared to Walk Up, compared to individuals aged >54 (OR = 2.88, CI = [1.85, 4.49], $p < .01$). Overall, Word of Mouth is more likely to be a source of study engagement for younger participants. Compared to those who reported living in San Ysidro, participants who lived outside of San Ysidro were 2.36 times more likely to report having heard of the testing site through their Clinic/Provider compared to Walk Up (OR = 2.36, CI = [1.85, 3.02], $p < .01$) and 2.11 times more likely to be recruited through Word of Mouth compared to Walk

**Table 2. Adjusted multinomial logistic regression model comparing predictors and source of recruitment information outcomes, compared to Walk Up.**

| Variable | Category | OR: Clinic/Provider | p | OR: Word of Mouth | p |
|---|---|---|---|---|---|
| **Sex** | | | | | |
| | Female | [reference] | | [reference] | |
| | Male | 0.65 (0.51, 0.83) | <.01 | 1.13 (0.90, 1.43) | .29 |
| **Ethnicity** | | | | | |
| | Hispanic | [reference] | | [reference] | |
| | Not Hispanic | 1.13 (0.70, 1.82) | .62 | 2.12 (1.37, 3.29) | <.01 |
| **Age** | | | | | |
| | >54 | [reference] | | [reference] | |
| | <18 | 1.93 (1.29, 2.88) | <.01 | 2.78 (1.87, 4.14) | <.01 |
| | 18-24 | 0.99 (0.60, 1.63) | .97 | 2.88 (1.85, 4.49) | <.01 |
| | 25-34 | 1.42 (0.99, 2.04) | .06 | 1.83 (1.28, 2.61) | <.01 |
| | 35-44 | 1.31 (0.89, 1.93) | .17 | 1.40 (0.95, 2.07) | .09 |
| | 45-54 | 0.96 (0.68, 1.34) | .80 | 1.00 (0.72, 1.42) | .96 |
| **City** | | | | | |
| | San Ysidro | [reference] | | [reference] | |
| | Outside SY | 2.36 (1.85, 3.02) | <.01 | 2.11 (1.66, 2.69) | <.01 |
| | Prefer not to say | 4.00 (1.55, 10.32) | <.01 | 3.29 (1.29, 8.38) | .01 |
| **Education** | | | | | |
| | High School | [reference] | | [reference] | |
| | No High School | 1.11 (0.76, 1.60) | .60 | 1.06 (0.74, 1.51) | .76 |
| | University | 1.40 (0.80, 2.45) | .24 | 1.37 (0.80, 2.35) | .26 |
| | Prefer not to say | 1.04 (0.77, 1.40) | .81 | 0.73 (0.55, 0.98) | .04 |
| **Symptoms** | | | | | |
| | Asymptomatic | [reference] | | [reference] | |
| | Symptomatic | 1.18 (0.93, 1.49) | .18 | 0.80 (0.63, 1.01) | .06 |

*Note:* Reference category is "Walk Up" for both Clinic/Provider and Word of Mouth comparisons. Statistics are formatted as Odds Ratio (OR) (95% Confidence Interval (CI)).

Up (OR = 2.11, CI = [1.66, 2.69], *p* < .01). Although significant in the chi-squared analysis, education and clinical symptoms were not significant in the full regression model. See Table 2 for more details.

## Discussion

Understanding patterns of successful participant engagement is essential to ensuring clinical and translational research participation among historically marginalized and underrepresented Hispanic/Latino(a) populations. Among the source of participant engagement identified and examined in this study, Word of Mouth (36%) emerged as the most frequently reported source of study engagement, followed by Clinic/Provider referrals (32%), and Walk Up (21%). These sources of participant engagement in our study were similar in their reliance on direct, in-person communication, which is consistent with several other studies [17–19]. This aligns with cultural values deeply ingrained in these communities including *personalismo* (emphasis on building relationships), *respeto* (respect), *familismo* (loyalty to family), and *confianza* (trust) [16,20]. These cultural principles emphasize the importance of direct, in-person communication as a vital approach for implementing effective and culturally meaningful outreach strategies to serve the needs of these communities. For instance, a community-based study employed in a Mexican-origin population, similar to the current study population,

discovered that face-to-face contact with community residents and partnerships with community-based organizations (CBOs) were most effective for the enrollment of participants in their study [17]. Passive recruitment methods/sources (e.g., flyers, word of mouth, and radio advertisements) were examined in previous research, with results emphasizing the greater odds of enrollment among Latino(a) immigrants in behavioral research given these recruitment sources [18]. These findings highlight the significance of Word of Mouth as an important source of recruitment, compared to Walking up, when engaging Latino(a) community members in public health research. Furthermore, our data reveal a notable gender difference in recruitment engagement, with male participants being less likely to report having heard of our testing site from a Clinic/Provider compared to female participants, particularly when compared to Walk Up. This trend may be attributable to the tendency for men to visit their healthcare clinic less frequently compared to women; studies have identified that gender disparity in healthcare, including primary care visits, is linked to factors including fear of diagnosis, mistrust in physicians, and adherence to traditional masculinity beliefs [21,22]. Consequently, men demonstrate lower levels of engagement with their primary care physicians, relative to women [23,24]. Our previous research highlights this trend by demonstrating that male participants were less likely to be vaccinated against COVID-19, suggesting a reduced use of primary and preventative healthcare services compared to female participants [14]. This finding offers insight into the sex gap observed in successful referrals among males and females, highlighting the importance of considering these differences when adopting specific recruitment strategies in future studies.

Sources of participant engagement also varied by age group within our study population. Younger participants (cohorts "<18" and "18-24"), were more than twice as likely to be recruited into our study through Word of Mouth than through Walk Up, compared to those >54 years of age. There are several factors that could contribute to this difference. For instance, younger age groups have been found to have more extensive social networks and engage more actively among a variety of communication platforms (e.g., social media and word of mouth) [25]. In addition, younger age groups may be more inclined to trust recommendations from their peers or family members, making Word of Mouth a more effective recruitment source among this age group compared to the older age groups who have been found to rely on more traditional sources, such as face-to-face interaction, for acquiring information [26]. A study conducted on the role of trust among young adolescents' decision to vaccinate against human papillomavirus (HPV) reported that their trust in their own mothers was stronger compared to other close/influential members [27]. This also highlights the importance of identifying influential members within social networks of various age groups [28]. By targeting referent individuals, enhanced recruitment and engagement strategies can be more effectively used in future community-based studies.

Leveraging interpersonal networks and healthcare provider relationships is critical, as suggested by our findings, particularly for engaging underserved communities in research, similar in composition to our study sample. Building and establishing trust within the community is also essential to enhancing recruitment and retention in research studies, especially among the Latino and Black communities [19]. Partnering with healthcare clinics and providers remains a notable recruitment strategy, due to the level of trust community members place in their healthcare network, as shown in our previous research [14]. Future studies should focus on identifying alternative recruitment strategies outside of the healthcare network to diversify outreach efforts and ensure representative study populations. Moreover, as the popularity and diversity of social media platforms increase, so does their utility for promoting clinical research [29]. Social media's increasing presence in our society for locating information demonstrates its potential as an innovative approach for increasing and engaging a broader audience [29]. Tailoring recruitment strategies to align with emerging communication platforms will be important for advancing public health research engagement. It is important to note that our sample was overwhelmingly recruited through traditional methods and sources. A source of participant engagement that did not provide many referrals in our study was the use of highway billboards, with just one participant reporting having heard of the study from a billboard. This finding, however, is not unique. A recruitment-focused study based in Baltimore found that billboards were one of the least effective and most expensive forms of recruitment [30]. Unsurprisingly, participants who lived in San Ysidro were significantly more likely to report Walk Up as their source of study engagement compared to living in a different city.

## Limitations and future directions

The primary limitation of our study is self-selection bias, meaning that the individuals represented in our sample are solely those who consented to be a part of our study, omitting individuals who chose to get tested at our site, but declined to participate and consent to the study; 60.8% of unique individuals who tested at our site agreed to also consent and participate in our research study. Further, our study's temporal context can be considered a limitation due to participant enrollment occurring throughout the height of the COVID-19 pandemic, and specifically for an infectious disease study. However, it is important to consider the value of these data gathered during an unprecedented time, a unique opportunity for similar time periods. Another limitation of this study is the generalizability of different geographical settings. While border communities may differ significantly between other cities, their importance and similarities have not yet been explored extensively. Lastly, the collection of study engagement was a component of the larger CO-CREATE project implemented during the peak of the COVID-19 pandemic, resulting in the lack of standardization during the collection process to minimize the burden imposed on study participants. Future studies should allow for multiple source responses, more specific responses (i.e., breaking down Word of Mouth into more specific categories), and the inclusion of qualitative data to provide context for quantitative results. This will allow for future research engagement efforts to be specifically tailored and culturally responsive for target study populations.

## Conclusion

Our study identified Walk Up, Clinic/Provider, and Word of Mouth as the three most successful sources of participant engagement in a COVID-19 testing research study in a U.S./Mexico border community. We observed varying differences in engagement by sex, ethnicity, age, city, education, and clinical symptoms common to historically marginalized and underrepresented groups in clinical and translational research. In order for public health research to become more representative and equitable, enhancing generalizability, future studies are needed to identify the most effective sources of successful participant engagement in diverse community contexts.

## Supporting information

**S1 File. Spreadsheet of de-identified raw data.**
(XLSX)

## Author contributions

**Conceptualization:** Angel Lomeli, Linda Salgin, Kelli L. Cain, Nicole A. Stadnick, Louise C. Laurent, Borsika A. Rabin, Marva Seifert.

**Data curation:** Angel Lomeli, Arleth A. Escoto, Breanna Reyes, Kayleigh Kornher, Keira Beltran-Murillo, Kathia Nuñez, Ariel Cohen, Maria Linda Burola, Isabel Villegas, Scarlet Flores, Ana Perez-Portillo, Norma Porras, Melody Torres.

**Formal analysis:** Angel Lomeli, Marva Seifert.

**Funding acquisition:** Louise C. Laurent.

**Investigation:** Angel Lomeli, Arleth A. Escoto, Breanna Reyes, Kayleigh Kornher, Keira Beltran-Murillo, Kathia Nuñez, Maria Linda Burola, Isabel Villegas, Scarlet Flores, Ana Perez-Portillo, Norma Porras, Linda Salgin, Nicole A. Stadnick, Louise C. Laurent, Borsika A. Rabin, Marva Seifert.

**Methodology:** Angel Lomeli, Maria Linda Burola, Linda Salgin, Kelli L. Cain, Nicole A. Stadnick, Louise C. Laurent, Borsika A. Rabin, Marva Seifert.

**Project administration:** Breanna Reyes, Maria Linda Burola, Ana Perez-Portillo, Linda Salgin, Kelli L. Cain, Nicole A. Stadnick, Louise C. Laurent, Borsika A. Rabin, Marva Seifert.

**Supervision:** Angel Lomeli, Breanna Reyes, Maria Linda Burola, Ana Perez-Portillo, Linda Salgin, Nicole A. Stadnick, Louise C. Laurent, Borsika A. Rabin, Marva Seifert.

**Validation:** Marva Seifert.

**Writing – original draft:** Angel Lomeli, Arleth A. Escoto, Breanna Reyes, Kayleigh Kornher, Keira Beltran-Murillo, Kathia Nuñez, Maria Linda Burola, Isabel Villegas, Scarlet Flores, Ana Perez-Portillo, Norma Porras, Melody Torres, Linda Salgin, Kelli L. Cain, Nicole A. Stadnick, Louise C. Laurent, Borsika A. Rabin, Marva Seifert.

**Writing – review & editing:** Angel Lomeli, Arleth A. Escoto, Breanna Reyes, Kayleigh Kornher, Keira Beltran-Murillo, Kathia Nuñez, Ariel Cohen, Maria Linda Burola, Isabel Villegas, Scarlet Flores, Ana Perez-Portillo, Norma Porras, Melody Torres, Linda Salgin, Kelli L. Cain, Nicole A. Stadnick, Louise C. Laurent, Borsika A. Rabin, Marva Seifert.

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
