## [Decision Letter · Decision Letter 0]

2 Jun 2025

PONE-D-25-11940Sources of Successful Participant Engagement in a Public Health Research Study: a Focus on a Latino CommunityPLOS ONE

Dear Dr. Lomeli,

Thank you for submitting your manuscript to PLOS ONE. After careful consideration, we feel that it has merit but does not fully meet PLOS ONE’s publication criteria as it currently stands. Therefore, we invite you to submit a revised version of the manuscript that addresses the points raised during the review process.

We look forward to receiving your revised manuscript.

Kind regards,

Taiwo Opeyemi Aremu

Academic Editor

PLOS ONE

3. Please expand the acronym “NIH/NIEHS” (as indicated in your financial disclosure) so that it states the name of your funders in full.

4. In the online submission form you indicate that your data is not available for proprietary reasons and have provided a contact point for accessing this data. Please note that your current contact point is a co-author on this manuscript. According to our Data Policy, the contact point must not be an author on the manuscript and must be an institutional contact, ideally not an individual. Please revise your data statement to a non-author institutional point of contact, such as a data access or ethics committee, and send this to us via return email. Please also include contact information for the third party organization, and please include the full citation of where the data can be found.

6. Please remove your figures from within your manuscript file, leaving only the individual TIFF/EPS image files, uploaded separately. These will be automatically included in the reviewers’ PDF.

Reviewers' comments:

Reviewer's Responses to Questions

**Comments to the Author**

1. Is the manuscript technically sound, and do the data support the conclusions?

Reviewer #1: Yes

Reviewer #2: Yes

2. Has the statistical analysis been performed appropriately and rigorously? 

Reviewer #1: Yes

Reviewer #2: Yes

3. Have the authors made all data underlying the findings in their manuscript fully available?

Reviewer #1: No

Reviewer #2: Yes

4. Is the manuscript presented in an intelligible fashion and written in standard English?

Reviewer #1: Yes

Reviewer #2: No

5. Review Comments to the Author

Reviewer #1: This is a well-written paper that presents important findings about engaging underrepresented communities in research.

A few points to consider and/or clarify:

1. Did participants know the CO-CREATE site was a research study or were they just looking for a COVID test? The study procedures section (top of p15 describes it at first as a project site but then item 7 refers to it as a study). It also seems to be referred to as a testing site throughout the paper. I can imagine people may have a different propensity to visit the site if they know it is specifically research (vs just looking for a COVID test) for the reasons described in the introduction, and this is really important for the fundamental research question around research engagement. It would also be helpful to present how many participants did not consent to the research, as this is mentioned as a limitation.

2. A little more detail would be helpful on the free response coding methods for participant engagement, how many people were doing the coding? Was it consensus-based, and how were discrepancies handled?

3. Where the regression is described in the methods section, it says Walk Up was the most common response but this contradicts Figure 1 indicating Word of Mouth.

4. In Figure 1, it would be better to reduce the dead space between the y-axis and start of bars so they're not floating so much. Also why is the total in Figure 1 different from the total in Table 1?

5. Was the chi-squared test run using the same level of granularity for each variable as listed in the table? If so, having that level of granularity in variables across 3 categories is bound to produce significance because there are so many combinations and cells that inevitably end up with very small numbers. Consider trying to combine where it makes sense conceptually (e.g., is there something about being from Tijuana vs Chula Vista that critically differentiates people and is important to capture or is it actually just important that they're not from SY?). I also see age wasn't tested, please explain why in the text (or if it was tested and the p-value is just missing because it is included in the regression, consider revisiting the granularity of that variable or testing as a continuous variable).

6. I can appreciate the rigorous modelling approach, but it doesn't seem to align with your stated research goal of characterizing sources of participant engagement. In addition to the characterization, it seems like you're also trying to make stronger statements about how those demographic characteristics interact with the engagement mechanism. For this type of research question and how the introduction is framed, it seems like it would be more meaningful to define participant archetypes/a set of characteristics simultaneously for each effective engagement approach rather than singularly like the regression is doing. A latent class analysis may work better for this and be more straightforward to interpret (and present for the reader, the regression results are a little hard to follow and quite cumbersome with all the levels, since the interpretation of any one variable is dependent not only on the reference for that variable but also all the other variables and respective references in the model, i.e. it's not just 'compared to X reference level' it's 'compared to X reference level when variables A, B, and C are also at X, Y, Z reference levels' which can become a bit unwieldy with a bunch of categorical covariates).

Reviewer #2: Although the study is generally understandable, there are places where descriptions are incomplete or not well explained. My attached file suggests a few of those modifications to improve the understanding of methods and results.

6. PLOS authors have the option to publish the peer review history of their article (what does this mean? ). If published, this will include your full peer review and any attached files.

**Do you want your identity to be public for this peer review?** For information about this choice, including consent withdrawal, please see our Privacy Policy .

Reviewer #1: No

Reviewer #2: **Yes: ** Dr. Alfredo L. Fort

---

## [Author Response · Author response to Decision Letter 1]

2 Sep 2025

Reviewer 1:

Did participants know the CO-CREATE site was a research study or were they just looking for a COVID test? The study procedures section (top of p15 describes it at first as a project site but then item 7 refers to it as a study). It also seems to be referred to as a testing site throughout the paper. I can imagine people may have a different propensity to visit the site if they know it is specifically research (vs just looking for a COVID test) for the reasons described in the introduction, and this is really important for the fundamental research question around research engagement. It would also be helpful to present how many participants did not consent to the research, as this is mentioned as a limitation:

Thank you for bringing this up. The testing site was advertised as a testing site first, and, upon testing, patients were offered participation in our study. While testing site and research study could both be used, since both are true, have modified the language when introducing this for clarity and transparency to the reader. We have also added the proportion of individuals who agreed to be a part of the study (60.8%) to the discussion section, as requested.

A little more detail would be helpful on the free response coding methods for participant engagement, how many people were doing the coding? Was it consensus-based, and how were discrepancies handled?:

This is a good suggestion – more information regarding how we came up with handling the free response engagement methods was added for transparency and replicability.

Where the regression is described in the methods section, it says Walk Up was the most common response but this contradicts Figure 1 indicating Word of Mouth:

Thank you for catching this, we have fixed the incorrect language in the text – Figure 1 is correct.

In Figure 1, it would be better to reduce the dead space between the y-axis and start of bars so they're not floating so much. Also why is the total in Figure 1 different from the total in Table 1?:

The different values in fig 1 and table 1 are due to table 1 describing the top three most common sources (rather than all of the sources identified). We have changed the title of the table to reflect this, since it was not clear before. Regarding the figure format, we agree that it looked awkward before due to the excessive white space – this has been fixed.

Was the chi-squared test run using the same level of granularity for each variable as listed in the table? If so, having that level of granularity in variables across 3 categories is bound to produce significance because there are so many combinations and cells that inevitably end up with very small numbers. Consider trying to combine where it makes sense conceptually (e.g., is there something about being from Tijuana vs Chula Vista that critically differentiates people and is important to capture or is it actually just important that they're not from SY?). I also see age wasn't tested, please explain why in the text (or if it was tested and the p-value is just missing because it is included in the regression, consider revisiting the granularity of that variable or testing as a continuous variable):

Yes, the chi-squares tests were run as shown in Table 1. Regarding age, it was run as a continuous variable at first (was significant at p<.01) but was changed to a categorical variable for ease of interpretability for the table and follow-up analysis. As a categorical variable, it was run and also significant at p <.01 but, as you mentioned, was not included in the Table – this was an error and has been fixed and added to the table. As suggested, city was consolidated into three categories – San Ysidro resident, not a San Ysidro resident, and prefer not to answer.

I can appreciate the rigorous modelling approach, but it doesn't seem to align with your stated research goal of characterizing sources of participant engagement. In addition to the characterization, it seems like you're also trying to make stronger statements about how those demographic characteristics interact with the engagement mechanism. For this type of research question and how the introduction is framed, it seems like it would be more meaningful to define participant archetypes/a set of characteristics simultaneously for each effective engagement approach rather than singularly like the regression is doing. A latent class analysis may work better for this and be more straightforward to interpret (and present for the reader, the regression results are a little hard to follow and quite cumbersome with all the levels, since the interpretation of any one variable is dependent not only on the reference for that variable but also all the other variables and respective references in the model, i.e. it's not just 'compared to X reference level' it's 'compared to X reference level when variables A, B, and C are also at X, Y, Z reference levels' which can become a bit unwieldy with a bunch of categorical covariates):

Thank you for this comment. We agree that a multinomial regression model as a final model might not fully capture specific subgroups that our research question is aiming to look at. We had originally considered a latent class analysis, among many different approaches, as a potential model to use for our analysis. Some other approaches we thought of included reducing the outcome variable to two levels (a more conventional categorical regression model) to make statistcal inferences less cumbersome – which would end up being Clinic/Provider vs. Word of Mouth, the two top methods. However, this would leave out the third most common choice which has a high frequency (495) – something we felt was important to include, especially given the next (fourth) highest response was at 78. When looking at the assumptions and model fit for an LCA, we found that the nature of our dataset showed that the model is not a good fit. For example, when calculating the Bayesian information criterion, we found that its value was lowest for the single class model and increased significantly and linearly as the classes increased from 2-5. Lastly, we have a lot of subgroups among each of the variables of interest (as you had pointed out in an earlier comment). Regarding consolidation, we were able to consolidate the city variable to be a comparison of living in San Ysidro vs. not in San Ysidro since, as you mentioned, the purpose of the study is able to be achieved with this consolidation. The chi-square and logistic regression model were re-run since this change would have changed the statistics – this is reflected in both tables in the manuscript.

Reviewer 2:

This might need a definition, to make it clearer for the reader:

Thank you for pointing this out, while this is explained in the main text, we see why it is not clear in the abstract – this has been rectified.

Where in? The one near San Diego, CA? Needs to be fully introduced.

It would be good to provide general socio-demographic characteristics of the city, so that the reader has an idea of the generic "context" of the study area:

More context and specific demographics were added in order to better introduce the region where the study took place, an important factor in introducing the study setting.

Put the term first in quotations, and then explain what they are in English, for the benefit of the reader:

This is a great point – we put promotores in italics, for uniformity across the paper since they were also used in the discussion when Spanish words were mentioned. We also slightly expanded on the role and model of the promotores for more context.

Important to describe where are the other areas...

Which type?

How far away, what characteristics of this place, etc., needed:

As suggested by another reviewer, we have revised the levels of the city variable to only include San Ysidro resident, non-San Ysidro resident, and prefer not to answer. However, we completely agree that it would be a good idea to include an explanation as to where our variable levels were collected including the possible symptoms that participants could have reported which would have resulted in them being classified as “symptomatic” in this study. This has been added to the methods section.

Important percentage; tricky that there is no answer:

If this is referring to the overall percentage of “prefer not to say”, for instance, they can be found in the fifth column under the total, along with their counts for each subgroup in each variable.

Needs to be put more clearer which is the OR, etc., as narrated in the text:

The title of each column was changed to reflect that they are presenting odds ratios. Also, for more clarity, the note under the table was changed to be more specific to what the comparison is.

compared to...?:

Thank you for catching this – this has been rectified.

Any reasons, given that there were open ended questions asked?:

We are unsure… our discussion after this sentence describes potential explanations backed by other studies, including one that precedes this one, who have shown a low rate of primary care and preventative care for men compared to women.

Also physical aspects such as being out and about at younger ages than at older ages...other possible factors?:

Perhaps, however, the study found that older age participants were actually more likely to walk up to our site, not the other way around.

---

## [Decision Letter · Decision Letter 1]

22 Sep 2025

Sources of successful participant engagement in a public health research study: a focus on a Latino community

PONE-D-25-11940R1

Dear Dr. Lomeli,

We’re pleased to inform you that your manuscript has been judged scientifically suitable for publication and will be formally accepted for publication once it meets all outstanding technical requirements.

Kind regards,

Taiwo Opeyemi Aremu, MD, MPH, PhD

Academic Editor

PLOS ONE

Additional Editor Comments (optional):

Reviewer #1:

Reviewer #2:

Reviewers' comments:

Reviewer's Responses to Questions

**Comments to the Author**

1. If the authors have adequately addressed your comments raised in a previous round of review and you feel that this manuscript is now acceptable for publication, you may indicate that here to bypass the “Comments to the Author” section, enter your conflict of interest statement in the “Confidential to Editor” section, and submit your "Accept" recommendation.

Reviewer #1: All comments have been addressed

Reviewer #2: All comments have been addressed

2. Is the manuscript technically sound, and do the data support the conclusions?

Reviewer #1: Yes

Reviewer #2: Yes

3. Has the statistical analysis been performed appropriately and rigorously? 

Reviewer #1: Yes

Reviewer #2: Yes

4. Have the authors made all data underlying the findings in their manuscript fully available?

Reviewer #1: Yes

Reviewer #2: Yes

5. Is the manuscript presented in an intelligible fashion and written in standard English?

Reviewer #1: Yes

Reviewer #2: Yes

6. Review Comments to the Author

Reviewer #1: Thank you for your thoughtful responses and edits. All of my comments have been addressed and I have no further notes.

Reviewer #2: Thanks for submitting the revised manuscript, improving several aspects for readers to easily understand the study and the study to be well and fully described. Please see the attached file for only a few final suggestions to clarify descriptions.

7. PLOS authors have the option to publish the peer review history of their article (what does this mean? ). If published, this will include your full peer review and any attached files.

**Do you want your identity to be public for this peer review?** For information about this choice, including consent withdrawal, please see our Privacy Policy .

Reviewer #1: No

Reviewer #2: **Yes: ** Dr. Alfredo L. Fort

---

## [Editor Report · Acceptance letter]

PONE-D-25-11940R1

PLOS ONE

Dear Dr. Lomeli,

I'm pleased to inform you that your manuscript has been deemed suitable for publication in PLOS ONE. Congratulations! Your manuscript is now being handed over to our production team.

Kind regards,

on behalf of

Dr. Taiwo Opeyemi Aremu

Academic Editor

PLOS ONE